# Absolute Risk and Attributable Fraction of Type-Specific Human Papillomavirus in Cervical Cancer and Precancerous Lesions—A Population-Based Study of 6286 Women in Rural Areas of China

**DOI:** 10.3390/jcm11216483

**Published:** 2022-10-31

**Authors:** Li Li, Mingyue Jiang, Tingyuan Li, Jian Yin, Ruimei Feng, Li Dong, Sufia Imam, Jianfeng Cui, Bin Liu, Xun Zhang, Qinjing Pan, Youlin Qiao, Wen Chen

**Affiliations:** 1Department of Clinical Research, The First Affiliated Hospital, Jinan University, Guangzhou 510000, China; 2Department of Cancer Epidemiology, National Cancer Center/National Clinical Research Center for Cancer/Cancer Hospital, Chinese Academy of Medical Sciences & Peking Union Medical College, Beijing 100021, China; 3School of Population Medicine and Public Health, Chinese Academy of Medical Sciences & Peking Union Medical College, Beijing 100021, China; 4Center for Cancer Prevention Research, Sichuan Cancer Hospital & Institute, Sichuan Cancer Center, School of Medicine, University of Electronic Science and Technology of China, Chengdu 610041, China; 5Department of Epidemiology and Health Statistics & Key Laboratory of Ministry of Education for Gastrointestinal Cancer, Fujian Medical University, Fuzhou 350122, China; 6Institutes of Biomedical Sciences, Shanxi University, Taiyuan 030006, China; 7International School, Jinan University, Guangzhou 510000, China

**Keywords:** cervical cancer, HPV genotype, HPV distribution, vaccine, cervical lesion

## Abstract

Background: To investigate the human papillomavirus (HPV) genotype distribution among the general population and assess the attribution of HPV genotypes targeted by vaccines to protect against cervical lesions theoretically. Methods: Cervical samples were collected from women aged 21 to 64 years old from Inner Mongolia and Shanxi Province in China who had not been vaccinated against HPV. HPV type-specific absolute risk (AR) to classified cervical lesions was calculated and then the attributable fraction (AF) was estimated, together with the combined contributions of the HPV types, targeted by four available HPV vaccines and five HPV vaccines in clinical trials in China to protect against cervical lesions. Results: A total of 6286 women with an average age of 44.1 years ± 8.41 (range: 21–64) participated in the study. The age distribution of 14 HR-HPV and HPV16/18 all showed a ‘U’ shape, which peaked in the ≤25 year-group and >55 year-group. The five most common genotypes were HPV16 (4.3%), HPV52 (4.1%), HPV58 (2.1%), HPV51 (2.1%), and HPV66 (1.7%). The prevalence of HPV types 6 and 11 infections was 1.1% and observed with *n* significant differences across age stratifications in China. AF to CIN2+ was predominated by HPV 16 with 56.2%, followed by HPV58 (12.0%), HPV52 (8.5%), HPV18 (4.3%), and HPV51 (2.9%). HPV52 and 58 in the prophylactic HPV vaccine would enhance the protection against CIN2+ by approximately 20%. Conclusions: Regarding multi-valent HPV vaccine development in China, the HPV types 16, 52, 58, and 18 should be given priority for their high prevalence at the population level, high AR, notable AF, and high relative risk to high-grade cervical lesions.

## 1. Introduction

A total of 570,000 new cases of cervical cancer worldwide was recently reported in 2018, ranking the fourth most prevalent disease in terms of both incidence and mortality among women [1]. Contrary to the decreasing trend in developed countries, incidence and mortality are still on the rise in China [2]. As per the latest data from China National Cancer Center, there were 111,000 new cases of cervical cancer (accounting for 5.6% of tumors in women) and 34,000 deaths in 2015, with the age of morbidity for cervical cancer becoming younger [2,3]. It is estimated that new cases of cervical cancer in China are likely to increase to 187,000 by 2050 if no effective measures are taken [4].

HPV is the common sexually transmitted virus worldwide. Infection of certain types of HPV is regarded as a causal and necessary factor for cervical cancer [5]. According to the report, the burden of HPV infection is significantly high, with about 10 million new cases of HSIL and 30 million new cases of LSIL being diagnosed every year [6]. More than 200 HPV genotypes have been identified and about 40 types of HPV have been found in the female genital tract [7]. According to their association with premalignant and malignant lesions, the HPVs are classified as high-risk HPV (HR-HPV) and low-risk HPV (LR-HPV). The prevalence and distributions of HPV genotypes vary with age, geographic locations, and economic development levels [7,8]. HPV infections precede cervical precancerous lesions and cervical cancer by 3–5 years [9]. Therefore, it is important to develop considerable knowledge about the HPV genotype distributions and change trends in different cervical lesions.

HPV-DNA-based screening and prophylactic vaccination are effective measures against cervical cancer [10,11]. The development of cervical cancer prevention strategies and the clinical management of CIN requires epidemiologic evidence of HPV prevalence and genotype distribution, which could be useful for the applications and developments of the screening program, prophylactic HPV vaccination and to guide the clinicians. A recent prediction model shows that if the two HPV infection prevention strategies (introducing vaccination for girls aged 12 years and expanding the population coverage of cervical cancer screening for women) were implemented, there would be an obvious decrease in the cervical cancer rate, and by the early 2070s, cervical cancer could be effectively eliminated [12]. Currently, four licensed prophylactic HPV vaccines have been approved by the National Medical Products Administration (NMPA) in China, including two bivalent (2Vhpv) vaccines targeted at HPV16 and 18, one quadrivalent HPV (4Vhpv) vaccine targeted at HPV6/11/16/18, and one nine-valent (9Vhpv) vaccine targeted at HPV6/11/16/18/31/33/45/52/58 [13,14]. As of October 2021, there are 32 types of Chinese vaccines under clinical evaluation. Of these, 10 candidates were in Phase III clinic trials [15]. 

In this study, we looked at the Absolute Risk (AR), Attribute Fraction (AF), and Relative Risk (RR) of single HPV genotypes, as well as the HPV infection status of different HPV vaccine groups, including both market and clinical trial vaccines, using different cytology and pathology stratification among rural women who had not been vaccinated against HPV. This work can illuminate the risk of different HPV infections and the variation trends among different HPV groups and vaccines in different disease status and provide evidence for efficient HPV vaccine designation and HPV prevention in the future.

## 2. Materials and Methods

### 2.1. Study Objects

The women from Inner Mongolia and Shanxi province in China who attended cervical cancer screening at local Women and Children’s hospitals were recruited from June 2016 to June 2017. Women were eligible if they (a) aged from 21 to 64 years; (b) had signed the informed consent form; (c) were not pregnant, or 8 weeks after the end of pregnancy; (d) had no history of hysterectomy; (e) were not menstruating when screening was done; (f) had not been vaccinated against HPV; and (g) had not undergone cervical cancer screening within three years. 

### 2.2. Demographic Information and Sample Collection 

The demographic information and risk factors were investigated by trained nurses with a structured questionnaire. The gynecologic examination was performed by experienced clinicians, and cervical samples were collected, kept in a PreservCyt^®^ medium, and stored under −80 °C.

### 2.3. HPV Tests

The first aliquot samples in Inner Mongolia were tested by cobas 4800 (Roche Diagnostics, GmbH, Mannheim, Germany), the samples from Shanxi Province were tested by Aptima (Hologic Inc., San Diego, CA, USA), and the rest were used for liquid-based cytology (LBC). The second aliquot was used for the HPV genotyping test (SPF10-LiPA system (INNO-LiPATM, Innogenetics, N.V., Ghent, Belgium), which uses a PCR-based hybridization line probe assay to detect 28 different HPV types, including 14 HR-HPV: HPV16/18/31/33/35/39/45/51/52/56/58/59/66/68; and 14 LR HPV: HPV6/11/26/40/43/44/53/54/69/70/71/73/74/82.

### 2.4. Cytology Diagnosis 

The LBC slides were reviewed by experienced cytologists at the National Cancer Center/Cancer Hospital Chinese Academy of Medical Sciences (CHCAMS), and the results were graded according to the Bethesda System [16]: negative for intraepithelial lesion or malignancy (NILM); atypical squamous cells of undetermined significance (ASC-US); atypical glandular cells (AGC); low-grade squamous intraepithelial lesions (LSIL); high-grade squamous intraepithelial lesions (HSIL); atypical squamous cells, not excluding HSIL (ASC-H); squamous cell carcinoma (SCC); adenocarcinoma in situ (AIS); and adenocarcinoma (ADC).

### 2.5. Colposcopy and Histological Diagnosis

HPV16/18-positive (cobas or Aptima tests) or ASCUS+ were referred to colposcopy within 12 weeks. Biopsies were taken directly on visual lesions, and random biopsies were taken if no visual lesions were detected for the subjects with cytology LSIL+. Histological diagnoses were performed by two experienced pathologists from CHCAMS and the diagnoses were categorized as: negative; cervical intraepithelial neoplasia grade 1 (CIN1); cervical intraepithelial neoplasia grade 2 (CIN2); cervical intraepithelial neoplasia grade 3 (CIN3); squamous cell carcinoma (SCC); adenocarcinoma in situ (AIS); and adenocarcinoma (ADC).

The final disease status for the women with no pathology results were classified in different pathology groups [17,18]: (1) Negative pathological diagnosis: (a) colposcopy results negative; (b) HPV DNA test negative and LBC diagnosed NILM or ASCUS; (c) HPV DNA test positive and LBC diagnosed NILM. (2) CIN2 group: cytology diagnosis was ASC-H, HSIL, AIS, or SCC. (3) No pathological results group: (a) HPV negative and LBC diagnosed Unsatisfactory, LSIL, or AGC; (b) HPV positive and LBC diagnosed ASCUS or LSIL.

### 2.6. Statistics Analysis

According to the HPV vaccines approved by NMPA or ongoing clinical trials in China, we categorized HPV infections into different groups: HPV6/11, HPV16/18, HPV16/18/58, HPV6/11/16/18, HPV16/18/52/58, other LR-HPV26/40/43/44/53/54/69/70/71/73/74/82, HPV 6/11/16/18/31/33/45/52/58, HPV6/11/16/18/31/33/45/52/58/59/68 and HPV6/11/16/18/31/33/35/39/45/51/52/56/58/59. Means and standard deviations were used to describe the numeric variable. HPV type-specific AR and corresponding AF [19] to different cervical lesions were calculated with 95% confidence interval. AR is the infection rate of HPV that does not distinguish multiple infections from single infections. The AF [20] was based on the relative number of instances in which each HPV infection type was observed as a single-type infection in a lesion of that grade and disease-type within each individual study. For example, in deriving an apportionment for two CIN 1 lesions found to test positive for both HPV 16 and 58 in a study, if there were 6 single-type HPV 16-infected CIN 1 lesions and 1 single-type HPV 58-infected lesion in that study, then [2 × 9/(9 + 1)] or 1.8 of these two multi-type-infected lesions would be attributed to HPV 16 and [2 × 9/(9 + 1)] or 0.2 attributed to HPV 58 [19]. The method used to calculate a confidence interval for a proportion is the Wilson score method without continuity correction. Two-sided confidence intervals for the single proportion: Comparison of seven methods [21]. Data were analyzed using SPSS 27.0 and EXCEL. Pearson’s χ^2^ test was used to analyze the significance of data. Statistical significance was set at *p* < 0.05 and the AF was calculated using a mathematical algorithm in EXCEL.

## 3. Results

### 3.1. Demographic Characteristics and HPV Distribution

In this study, 6241 women were screened and 6286 (97.9%) women had met the inclusion criteria. Table 1 shows the demographics of the total 6286 women (Mean age: 44.1 years ± 8.41). The women who did not have a drinking habit were less likely to be HPV-positive than the women who had a drinking habit (HPV: χ^2^ = 5.20, *p* = 0.023; HR-HPV: χ^2^ = 4.83, *p* = 0.028). The HPV-positive rates were different among women using different contraceptive methods (HPV: χ^2^ = 9.37, *p* = 0.009; HR-HPV: χ^2^ = 9.62, *p* = 0.008). The HPV-positive rates in women using condoms were significantly lower than women without any contraception methods (HPV: χ^2^ = 5.91, *p* = 0.015; HR-HPV: χ^2^ = 5.88, *p* = 0.015) and women who used precautions other than condoms (HPV: χ^2^ = 9.33, *p* = 0.002; HR-HPV: χ^2^ = 9.62, *p* = 0.002).

The age distribution of HPV infection by cytological and pathological grade is shown in Figure 1. The curves of the positive rates for total HPV (28 types), 14 HR-HPV, and HPV16/18 all showed a ‘U’ shape, which peaked in the ≤25-year group and >55-year group (Figure 1A). The prevalence of women with cytological HSIL increased with age and accounted for the highest age in 46–50 age group (Figure 1B). The prevalence of pathological CIN2 accounted for the highest age in the 41–45 age group, and for the CIN3+ lesions, the summit age group was 46–50 (Figure 1C).

### 3.2. AR, AF, and RR in Different Cytological and Pathological Grade

The five most common genotypes were HPV 16, 52, 51, 58, and 6616 (4.3%), HPV52 (4.1%), HPV58 (2.1%), HPV51 (2.1%), and HPV66 (1.7%); In women with NILM: the top five types of HPV infections indicated by AR or AF were HPV52, 16, 51, 66, and 58. Among 223 LSIL women: HPV16, 52, 56, 31, and 58 were the most frequent HPV types by the order of AR, which is a little different from the AF, the infection rate of HPV 58 increased from the fifth to the third, and the HPV 56 infection decreased from the third to the fourth; For the RR, HPV52 was the highest, followed by HPV58, 66, 16, and 18. In 147 HSIL or above women, the AR of HPV16, 58, 52, 18, and 51 were higher than the other HPV types, which is the same for the AF ranks; The RR ranks, from high to low, were HPV16, 31, 58, 18, and 33, respectively (Table 2).

The HPV types with a higher AR in CIN1 were: HPV58, 16, 52, 18, 33 and 51, and the AF were HPV58, 16, 52, 33, 18, the order of HPV33 and HPV18 were changed; For the RR, the HPV58 was the highest, followed by HPV33, 16, 18, 52. By contrast, the 5 HPV types with a higher AR in CIN2+ were: HPV16, 52, 58, 18, 51, 35; For AF, the types were: HPV16, 58, 52, 18 and 51, the order of HPV52 and HPV 58 were changed; The RR ranks, from high to low, were HPV16, 58, 18, 33, 45, respectively (Table 3).

### 3.3. Estimating the Impact of Different HPV Infection Groups and Vaccines in Different Cytological and Pathological Grades

HPV prevalence was 24.2% for 28 HPV types, 19.3% for 14 HR-HPV, and 1.1% for HPV 6/11. The cytological diagnosis was as below: 5309 (84.5%) NILM, 876 (13.9%) abnormal cytology (ASCUS or worse), 101 (1.6%) unsatisfactory. The prevalence of different HPV infection groups in 6286 women, stratified by cytology, is shown in Table 4. The HPV positivity rate increased with the elevation of cytological grades in 14 HR-HPV and 8 HPV vaccine groups (*p* < 0.001). The HPV prevalence in the groups of NILM and abnormalities were 49 (0.9%) and 19 (2.1%) for HPV6/11, and 72 (14.5) and 421 (48.1%) for 14 HR-HPV. The other LR-HPV were 300 (5.7%) and 118 (28.2%), HPV16/18 were 181 (3.4%) and 161 (47.1%), HPV6/11/16/18 were 229 (4.3%) and 176 (43.5%), HPV6/11/16/18/31/33/45/52/58 were 536 (10.1%) and 321 (38.2%), HPV16/18/58 were 238 (4.5%) and 223 (25.5%), HPV16/18/52/58 were 399 (7.5%), HPV6/11/16/18/31/33/45/52/58/59/68 were 607 (11.4%) and 350 (40.0%), and HPV6/11/16/18/31/33/35/39/45/51/52/56/58/59 were 724 (13.6%) and 401 (45.8%), respectively.

For all the pathology results: 5861 (93.2%) normal, 98 (1.6%) CIN1, 68 (1.1%) CIN2, 59 (0.9%) CIN3, 5 (0.01%, 79.54 per 100 000) cancers, and 195 (3.1%) had no results. Table 5 describes the prevalence of different HPV infection groups in 6286 women, stratified by pathology. The HPV infection rates of the 14 HR-HPV and 8 HPV vaccine groups also increased with pathological grade (*p* < 0.001). In the group of HPV 6/11 and other LR-HPV, there were no obvious linear trends with the pathological grade (*p* > 0.005). In the pathological normal and CIN2+ cases, the prevalence in the HPV6/11 group was 61 (1.0%) and 2 (1.5%), 923 (15.7%) and 119 (90.2%) in the 14 HR-HPV group, 362 (6.2%) and 17 (12.9%) in the other LR-HPV group, 220 (3.8%) and 83 (62.3%) in HPV16/18, 279 (4.8%) and 83 (62.3%) in HPV6/11/16/18, and 639 (10.9%) and 114 (86.4%) in HPV6/11/16/18/31/33/45/52/58, 294 (5.0%) and 98 (74.2%) in HPV16/18/58, 480 (8.2%) and 108 (81.8%) in HPV16/18/52/58, 722 (12.3%) and 114 (86.4%) in HPV6/11/16/18/31/33/45/52/58/59/68, and 868 (14.8%) and 118 (89.4%) in HPV6/11/16/18/31/33/35/39/45/51/52/56/58/59, respectively. From Table 4 and Table 5, we found that the change in the trend of LR-HPV was different with HR-HPV.

## 4. Discussion

This cross-sectional study mainly discussed the HPV prevalence and genotype distribution in different age groups and cervical lesions. We focused on women without HPV vaccination to elaborate the level of HPV infection in China, evaluate the oncogenic potential of individual HPV genotypes, and estimate the population-based impact of 4 HPV vaccines approved by NMPA and 5 HPV vaccines in clinical trial in China.

The prevalence of overall HPV infection and 14 HR-HPV infections had two peaks at age of <25 years old and at an age of >55 years old (Figure 1), similar results were also shown in other studies [22,23,24,25,26]. The 14 HR-HPV infection rate was 14.5% among women with normal cytology and 48.1% for abnormal cytology, which has also been similarly reported among other populations in China [22,27]. The abnormal cytology and pathology cases were mainly distributed in the 41–50 age group, which indicates that we need to take some preventive measures, such as strengthening screening for women aged 41–50.

HR-HPV prevalence is 19.3% and 5 (79.54 per 100,000) cervical cancer cases were found in this screening program. Both the HR-HPV infection rate and cervical cancer prevalence were higher in women from Shanxi and Inner Mongolia (our data) than in Shanghai (14.5%, 8.85 per 100,000) [28], Gansu (14.8%, 29.46 per 100,000) [29,30], and Yunnan (12.8%, 21.21 per 100,000). The higher incidence of HPV in this study may be attributed to several factors, such as higher proportion of old women (age of 50 years above women were 44.8%), rare use of condoms (most rural women used IUD, tubal ligation and oral contraceptive drugs for contraception), or other high-risk behaviors, e.g., coitus interruptus and fertile period contraception. Moreover, another important reason was the higher sensitivity of the detection methods used for HPV genotyping. The SPF10-lipa used for HPV genotyping in our current study was more sensitive than other HPV detection methods and widespread applications in the clinical trials of HPV vaccine [31]. The high cervical lesion prevalence was due to two factors: Firstly, unlike the traditional screening method-Pap smear, we used HPV and cytology combined screening in this study, and any positive results were referred for colposcopy, which can effectively improve the detection rate of cervical lesions. Second, all the cytology results were reviewed by experienced cytologists.

For women with normal cytology, the most common HPV types were similar to other studies in other areas of China and Asian area. HPV 16 was the most common type in our study for abnormal cytology women, which can be attributed to 12.7% of LSIL cases, 44.3% of HSIL+, and 56.2% of CIN2+ [20,23]. In the CIN1 cases, the most common HPV types were HPV 58, followed by HPV 16, and HPV 52, which was also reported in the Yangtze River Delta area, China [32]; however, the most common types were HPV 52, followed by HPV 16 and 58 in Taizhou, China [22]. HPV 16, 58, and 52 ranked the top three high-risk HPV types for CIN, which was consistent with the study in China [33]. Among the HSIL+, CIN1, and CIN2+ cases, HPV 18 was the fourth most common type in our study; however, many studies showed that HPV 18 was strongly associated with ADC [34,35]. For the RR, we found that over the severity of cervical lesions, the RR for the HR-HPV increased obviously. HPV16, 58 and 18 were the top three HPV types followed by HPV33, 45 and 52. HPV52 was the most common HPV type in China, and while it did not present a high RR in our study, its high AR and high AF cannot be ignored. HPV testing techniques should focus on these types herein. Most HPV tests for cervical cancer screening include 14 HR-HPV at present, which leads to a high sensitivity for detecting CIN2+ and a substantial loss in specificity [36]. Therefore, HPV tests require additional triage to stratify HPV screen-positive women according to risks for CIN2+ [37] since the relative risk and attribute fraction varied considerately by the individual genotype of 14 HR-HPV. Consequently, HPV genotyping could serve as a triage to assess an HPV-positive woman’s risk more accurately. Considering the importance of 52 and 58 in CIN2+, future HPV-based cervical cancer screening technology may need to genotype these two strains.

Genital warts is the second most common viral sexually transmitted infection in China [38]. HPV 6 and 11 were the most common LR HPV types causing genital warts and contribute to more than 90% of condyloma [39,40,41,42]. According to previous reports [43], the LR HPV-positive rate was highest in lower vagina samples and cervical samples and HPV 6 and 11 infections in vagina were 1.71% to 3.4% in China. Our study also observed the prevalence of HPV 6 and 11 infections in the cervix (1.10% taken together); however, there was no significant change across the different age groups. It has been approved that an MSD tetravalent HPV vaccine could not achieve a clinical endpoint for genital warts due to the low infection rates of HPV 6 and 11 and the low prevalence of genital warts in the Chinese population [44]. Thus, future clinical trial designers of the vaccines should pay attention to this point.

Research shows that the bivalent HPV vaccine, quadrivalent HPV vaccine, and nine-valent HPV vaccine have the potential to prevent approximately 70–90% of cervical cancer cases [45,46]. Recent studies have depicted the HPV prevalence status in China regarding the different HPV types, but few studies have shown the infection status of the HPV groups. Estimating the pre-immunization prevalence of HPV infection and the distribution of HPV types is fundamental to understanding the subsequent impact of HPV vaccination and HPV screening. Our study produced good data regarding HPV prevalence by different combinations of 9 HPV vaccine groups in China: For the vaccines available in China, the combined contribution of HPV 16/18 in bivalent vaccine and HPV 6/11/16/18 in quadrivalent vaccine were 62.3% to CIN2+ and 100% to cervical cancer; for the 9-valent vaccines targeted at HPV6/11/16/18/31/33/45/52/58, the combined contribution was 86.4% to CIN2+, which suggests that the bivalent vaccine and quadrivalent vaccine would provide approximately 62.3% and 86.4% protection, respectively, against CIN2+ in China. For the vaccines in clinical trials in China, a trivalent vaccine targeted at HPV16/18/58 would provide approximately 74.2% protection against CIN2+. Compared to a bivalent vaccine, the addition of HPV58 to a prophylactic HPV vaccine would increase protection against CIN2+ by approximately 12%. The quadrivalent vaccine targeted at HPV16/18/52/58 would increase protection against CIN2+ by approximately 20% with the addition of HPV52 and 58. The higher contributions of HPV52 and HPV58 indicate that it will be valuable for clinic trial design and estimation of the vaccines’ potential protection. Also, for ASCUS+ women, the combined contribution of carcinogenic HPV types were: bivalent vaccines targeted at HPV16/18 (18.4%), 9-valent vaccine targeted at HPV6/11/16/18/31/33/45/52/58 (36.6%), trivalent vaccine targeted at HPV16/18/58 (25.5%), quadrivalent vaccines targeted at HPV16/18/52/58 (32.2%), 11-valent vaccine targeted at HPV6/11/16/18/31/33/45/52/58/59/68 (40.0%), and 14-valent vaccine targeted at HPV6/11/16/18/31/33/35/39/45/51/52/56/58/59 (45.8%). According to the guidelines of cervical cancer prevention and control in China, ASCUS with HPV-positive or cytology diagnosed as ASC-H or above should be referred for colposcopy, HPV vaccination can effectively reduce colposcopy referral rates.

The limitations of our study were: (1) The smoking rates among Chinese women are low. In our study, the smoking rate is too small to evaluate its role in HPV infection. (2) We only have the data about women in the Shanxi and Inner Mongolia provinces; two regions with high HPV prevalence areas. But all the women in our study came from the general population and didn’t undergo cervical cancer screening within recent three years, so it can truly reflect the prevalence of HPV infection in China. In addition, the cytology and pathology results were graded by senior doctors in CICAMS and reported an HPV prevalence of LR-HPV infection, which was seen in only a few studies.

## 5. Conclusions

Our study suggested that the type-specific prevalence of HPV changes with age, and HPV16, 18, 52, and 58 had high prevalence, absolute risk, notable attributable fraction, and relative risk to high-grade cervical lesions among Chinese women. The priority should be given to HPV16, 52, 58, and 18 for multi-valent HPV vaccine development and HPV screening programs in China.

## Figures and Tables

**Figure 1 jcm-11-06483-f001:**
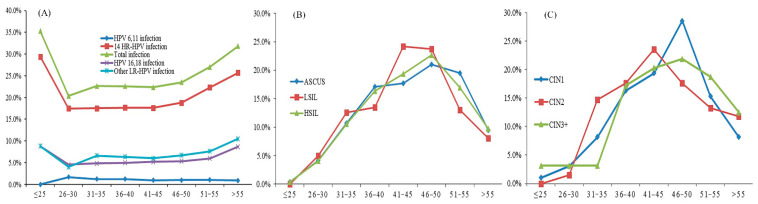
(**A**) The HPV infection rate, (**B**) cytology distribution, and (**C**) pathology distribution in different age groups.

**Table 1 jcm-11-06483-t001:** Characteristics and distribution of HPV infection among women living in China *n* = 6286.

Characteristics	Number (%)	HPV ^⁑^ Infection (%)	*p* Value	14HR-HPV Infection (%)	*p* Value	
Age ^※^						
21–29	383 (6.1)	83 (21.7)	0.004	71 (18.5)	0.006	
30–39	724 (11.5)	164 (22.7)		127 (17.5)		
40–49	2364 (37.6)	531 (22.5)		417 (17.6)		
Above 50	2815 (44.8)	741 (26.3)		601 (21.3)		
Mean ± SD	44.05 ± 8.412					
Number of pregnancy *						
0	54 (0.9)	12 (22.2)	0.118	10 (18.5)	0.067	
1–2	2287 (36.4)	514 (22.5)		406 (17.8)		
3–4	3153 (50.2)	789 (25.0)		645 (20.5)		
≥5	788 (12.5)	202 (25.6)		154 (19.5)		
Number of children ^#^						
0	92 (1.5)	23 (25.0)	0.130	21 (22.8)	0.073	
1–2	4979 (79.3)	1170 (23.5)		935 (18.8)		
3–4	1133 (18.1)	301 (26.6)		240 (21.2)		
≥5	73 (1.2)	21 (28.8)		18 (24.7)		
Education						
Illiterate	535 (0.8)	116 (21.7)	0.168	93 (17.4)	0.237	
Literate	5751 (91.5)	1430 (21.4)		1123 (19.5)		
Education level						
Primary school	1245 (21.6)	282 (22.7)	0.020	217 (17.4)	0.392	
Junior high school	2262 (39.3)	564 (24.9)		445 (19.7)		
High school	840 (14.6)	235 (28.0)		208 (24.8)		
University or above	1404 (24.4)	322 (22.9)		253 (18.0)		
Alcohol use						
Never	5090 (81.0)	1199 (23.6)	0.023	957 (18.8)	0.028	
Ever	1195 (19.0)	319 (26.7)		258 (21.6)		
Smoking						
Never	6139 (97.7)	1489 (24.3)	0.282	1193 (19.4)	0.251	
Ever	147 (2.3)	30 (20.4)		23 (15.6)		
Contraceptive method						
No contraception	1865 (29.7)	480 (25.7)	0.008	389 (20.9)	0.009	
Condom	945 (15.0)	194 (20.5)		151 (16.0)		
Others ^⁜^	3476 (55.3)	845 (24.3)		676 (19.4)		

⁑ human papillomavirus (HPV); high-risk HPV (HR-HPV); ^※^ years; * 4 data missing; ^#^ 9 data missing; ^⁜^ IUD, tubal ligation and oral contraceptive drugs.

**Table 2 jcm-11-06483-t002:** The absolute risk and attributable proportion of different HPV types by cytology.

HPV Type	NILM		LSIL		Relative Risk	HSIL+		Relative Risk
	Absolute Risk (%)	Attribute Fraction (%)	Absolute Risk (%)	Attribute Fraction (%)		Absolute Risk (%)	Attribute Fraction (%)	
HPV6	0.7 (0.5–0.1)	0.5 (0.4–0.8)	2.8 (1.3–6.1)	0	0	0.7 (0.2–3.8)	0	0
HPV11	0.2 (0.1–0.4)	0.1 (0.1–0.3)	1.4 (0.5–4.0)	0	0	0.7 (0.2–3.8)	0	0
HPV16	2.6 (2.2–3.0)	2.1 (1.8–2.5)	16.6 (12.3–22.0)	12.7 (8.9–17.7)	6.0	46.3 (38.4–54.3)	44.3 (36.5–52.4)	21.1
HPV18	1.0 (0.7–1.3)	0.7 (0.5–1.0)	5.4 (3.10–9.17)	3.2 (1.5–6.4)	4.6	8.8 (5.2–14.5)	5.6 (2.9–10.6)	8.0
HPV31	1.1 (0.8–1.4)	0.1 (0.1–0.3)	10.0 (4.5–8.06)	3.5 (1.8–6.9)	35	2.0 (0.7–5.8)	2.0 (0.7–5.8)	20
HPV33	0.6 (0.4–0.8)	0.4 (0.3–0.7)	1.8 (0.70–4.52)	0.6 (0.12–2.69)	1.5	4.1 (1.9–8.6)	2.5 (0.9–6.5)	6.3
HPV35	0.5 (0.4–0.8)	0.3 (0.2–0.5)	3.6 (1.83–6.92)	1.2 (0.37–3.62)	4.0	0	0	0
HPV39	1.1 (0.8–1.4)	0.8 (0.6–1.1)	6.3 (3.78–10.26)	3.8 (1.9–7.1)	4.8	2.1 (0.7–6.2)	0	0
HPV45	0.4 (0.3–0.6)	0.3 (0.2–0.5)	0.9 (0.25–3.21)	0.5 (0.1–2.5)	1.7	0	0	0
HPV51	1.7 (1.4–2.1)	1.3 (1.0–1.6)	7.2 (4.46–11.34)	5.8 (3.4–9.7)	4.5	6.1 (3.3–11.2)	3.0 (1.2–7.1)	2.3
HPV52	3.2 (2.8–3.8)	2.8 (2.4–3.3)	13.9 (9.97–19.06)	10.1 (6.8–14.8)	3.6	14.3 (9.5–20.6)	8.6 (5.1–14.3)	3.1
HPV56	1.0 (0.8–1.3)	0.7 (0.5–1.0)	11.7 (8.08–16.54)	8.0 (5.1–12.3)	11.4	0.7 (0.1–3.9)	0	0
HPV58	1.2 (1.0–1.6)	1.0 (0.7–1.3)	9.4 (6.24–13.97)	8.5 (5.5–12.9)	8.5	16.3 (11.2–23.1)	14.8 (10.0–21.5)	14.8
HPV59	0.6 (0.4–0.9)	0.5 (0.3–0.7)	3.6 (1.83–6.92)	2.5 (1.2–5.6)	5.0	0.7 (0.1–3.9)	0	0
HPV66	1.4 (1.1–1.8)	1.0 (0.7–1.3)	8.1 (5.17–12.40)	6.9 (4.3–11.0)	6.9	1.4 (0.4–4.8)	0	0
HPV68	1.1 (0.8–1.4)	0.8 (0.6–1.1)	4.0 (2.14–7.49)	1.3 (0.4–3.8)	1.6	2.0 (0.7–5.8)	1.4 (0.4–4.9)	1.8

Note: The figures in the brackets represent 95% confidence interval; negative for intraepithelial lesion or malignancy (NILM); low-grade squamous intraepithelial lesions (LSIL); high-grade squamous intraepithelial lesions or above (HSIL+).

**Table 3 jcm-11-06483-t003:** The absolute risk and attributable proportion of different HPV types by pathology.

HPV Type	Negative		CIN1		Relative Risk	CIN2+		Relative Risk
	Absolute Risk (%)	Attributable Fraction (%)	Absolute Risk (%)	Attributable Fraction (%)		Absolute Risk (%)	Attributable Fraction (%)	
HPV6	0.8 (0.6–1.0)	0.6 (0.4–0.8)	3.1 (1.1–8.7)	1.3 (0.3–6.1)	2.2	0.8 (0.1–4.2)	0	2.2
HPV11	0.3 (0.2–0.4)	0.1 (0.1–0.3)	0.0	0	0	0.8 (0.1–4.2)	0	0
HPV16	2.8 (2.4–3.3)	2.6 (2.2–3.0)	20.8 (14.0–24.0)	17.4 (11.2–26.1)	6.7	57.6 (40.1–65.7)	56.2 (47.7–64.4)	21.6
HPV18	1.1 (0.9–1.4)	0.8 (0.6–1.1)	8.3 (4.3–15.5)	4.6 (1.9–10.7)	5.8	6.1 (3.1–11.5)	4.3 (1.9–9.2)	5.4
HPV31	1.2 (0.9–1.5)	0.9 (0.7–1.2)	6.3 (2.9–12.9)	2.3 (0.7–7.6)	2.6	1.5 (0.4–5.4)	1.1 (0.3–4.8)	1.2
HPV33	0.6 (0.4–0.8)	0.5 (0.3–0.7)	7.3 (3.6–14.2)	5.2 (2.3–11.5)	10.4	3.8 (1.6–8.6)	2.2 (0.7–6.3)	4.4
HPV35	0.6 (0.4–0.8)	0.4 (0.2–0.6)	3.1 (1.1–8.7)	1.2 (0.2–5.8)	3.0	4.6 (2.1–9.6)	0	0
HPV39	1.2 (0.9–1.5)	0.9 (0.7–1.2)	1.0 (0.2–5.6)	0	0	3.0 (1.2–7.5)	0	0
HPV45	0.4 (0.2–0.6)	0.3 (0.2–0.5)	2.1 (0.6–7.2)	0	0	0.8 (0.1–4.2)	0.8 (0.1–4.2)	2.7
HPV51	1.8 (1.5–2.1)	1.3 (1.1–1.7)	7.3 (3.6–14.2)	4.3 (1.7–10.3)	3.3	6.1 (3.1–11.5)	2.9 (1.1–7.4)	2.2
HPV52	3.4 (3.0–3.9)	3.3 (2.8–3.7)	16.7 (10.6–25.3)	13.1 (7.8–21.2)	4.0	14.4 (9.4–21.4)	8.5 (4.8–14.5)	2.6
HPV56	1.3 (1.0–1.6)	0.9 (0.7–1.2)	6.3 (2.9–12.9)	0	0	3.0 (1.2–7.5)	0	0
HPV58	1.5 (1.2–1.8)	1.1 (0.9–1.4)	21.9 (14.8–31.0)	19.4 (12.8–28.3)	17.6	12.1 (7.6–18.8)	12.0 (7.5–19.0)	10.9
HPV59	0.7 (0.5–0.9)	0.5 (0.3–0.7)	2.1 (0.6–7.2)	0	0	0.8 (0.1–4.2)	0	0
HPV66	1.5 (1.2–1.8)	1.0 (0.8–1.3)	3.1 (1.1–8.7)	3.1 (1.1–8.6)	3.1	3.8 (1.6–8.6)	1.2 (0.3–4.9)	1.2
HPV68	1.1 (0.9–1.4)	0.8 (0.6–1.1)	2.1 (0.6–7.2)	0	0	1.5 (0.4–5.7)	0	0

Note: The figures in the brackets represent 95% confidence interval; cervical intraepithelial neoplasia grade 1 (CIN1); cervical intraepithelial neoplasia grade 2 or above (CIN2+).

**Table 4 jcm-11-06483-t004:** The prevalence of different HPV type combinations in 6286 women, China, stratified by cytology *n* (%).

Genotyping	Total * (*n* = 6185)	NILM (*n* = 5309)	ASCUS (*n* = 486)	LSIL (*n* = 223)	AGC (*n* = 20)	HSIL (*n* = 134)	Cancer (*n* = 13)	ASCUS+ (*n* = 876)
HPV6/11	68 (1.1)	49 (0.9)	8 (1.6)	9 (4.0)	0 (0.0)	1 (0.7)	1 (7.7)	19 (2.1)
14 HR-HPV	1193 (19.3)	772 (14.5)	125 (25.7)	162 (72.6)	8 (40.0)	113 (84.3)	13 (100.0)	421 (48.1)
Other LR-HPV 26/40/43/44/53/54/69/70/71/73/74/82	418 (6.8)	300 (5.7)	44 (9.1)	53 (23.8)	1 (5.0)	19 (14.2)	1 (7.7)	118 (13.5)
Vaccines approved by NMPA								
HPV16/18	342 (5.5)	181 (3.4)	33 (6.8)	47 (21.1)	2 (10.0)	69 (51.5)	10 (76.9)	161 (18.4)
HPV6/11/16/18	405 (6.5)	229 (4.3)	41 (8.4)	54 (24.2)	2 (10.0)	69 (51.5)	10 (76.9)	176 (20.1)
HPV6/11/16/18/31/33/45/52/58	867 (14.0)	536 (10.1)	102 (21.0)	105 (47.1)	7 (35.0)	105 (78.4)	12 (92.3)	321 (36.6)
Vaccines in clinical trial ^#^								
HPV16/18/58 ①	461 (7.5)	238 (4.5)	54 (11.1)	66 (29.6)	4 (20.0)	88 (65.7)	11 (84.6)	223 (25.5)
HPV16/18/52/58 ②	681 (11.0)	399 (7.5)	80 (16.5)	87 (39.0)	5 (25.0)	99 (73.9)	11 (84.6)	282 (32.2)
HPV6/11/16/18/31/33/45/52/58/59/68 ③	957 (15.5)	607 (11.4)	111 (22.8)	113 (50.7)	7 (35.0)	107 (79.9)	12 (92.3)	350 (40.0)
HPV6/11/16/18/31/33/35/39/45/51/52/56/58/59 ④	1125 (18.2)	724 (13.6)	121 (24.9)	148 (66.4)	8 (40.0)	111 (82.8)	13 (100.0)	401 (45.8)

* A total number of 6,286,101 dissatisfied cytology cases were excluded; negative for intraepithelial lesion or malignancy (NILM); atypical squamous cells of undetermined significance (ASCUS); atypical glandular cells (AGC); low-grade squamous intraepithelial lesions (LSIL); high-grade squamous intraepithelial lesions (HSIL); ASCUS or above(ASCUS+); National Medical Products Administration (NMPA). ^#^ The vaccines in clinical trial for the HPV types, which were the same as four Vaccines approved by NMPA, were excluded. ① The 3-valent vaccine from Health Guard Biological Technology Inc. (Hong Kong, China). ② The 4-valent vaccine from SL PHARM Inc. (Beijing, China). ③ The 11-valent vaccine from National Vaccine & Serum Institute (Beijing, China). ④ The 14-valent vaccine from Sinocelltech Ltd. (Beijing, China).

**Table 5 jcm-11-06483-t005:** The prevalence of different HPV types combination in 6286 women, China, stratified by pathology *n* (%).

Genotyping	Total ^#^ (*n* = 6091)	Negative (*n* = 5861)	CIN1 (*n* = 98)	CIN2 (*n* = 68)	CIN3 (*n* = 59)	Cancer (*n* = 5)	CIN2+ (*n* = 132)
HPV6/11	66 (1.1)	61 (1.0)	3 (3.1)	1 (1.5)	0 (0.0)	1 (20.0)	2 (1.5)
14 HR-HPV	1117 (18.3)	923 (15.7)	75 (76.5)	58 (85.3)	56 (94.9)	5 (100.0)	119 (90.2)
Other LR-HPV26/40/43/44/53/54/69/70/71/73/74/82	399 (6.6)	362 (6.2)	20 (20.4)	12 (17.6)	4 (6.8)	1 (20.0)	17 (12.9)
Vaccines approved by NMPA							
HPV16/18	331 (5.4)	220 (3.8)	28 (28.6)	37 (54.4)	41 (69.5)	5 (100.0)	83 (62.3)
HPV6/11/16/18	392 (6.4)	279 (4.8)	30 (30.6)	37 (54.4)	41 (69.5)	5 (100.0)	83 (62.3)
HPV6/11/16/18/31/33/45/52/58	818 (13.4)	639 (10.9)	65 (66.3)	55 (80.9)	54 (91.5)	5 (100.0)	114 (86.4)
Vaccines in clinical trial ^#^							
HPV16/18/58	438 (7.2)	294 (5.0)	46 (46.9)	47 (69.1)	46 (78.0)	5 (100.0)	98 (74.2)
HPV16/18/52/58	644 (10.6)	480 (8.2)	56 (57.1)	52 (76.5)	51 (86.4)	5 (100.0)	108 (81.8)
HPV6/11/16/18/31/33/45/52/58/59/68	901 (14.8)	722 (12.3)	65 (66.3)	55 (80.9)	54 (91.5)	5 (100.0)	114 (86.4)
HPV6/11/16/18/31/33/35/39/45/51/52/56/58/59	1059 (17.4)	868 (14.8)	73 (74.5)	57 (83.8)	56 (94.9)	5 (100.0)	118 (89.4)

^#^ A total number of 6,286,285 cases without pathological data were excluded; cervical intraepithelial neoplasia grade 2 (CIN2); cervical intraepithelial neoplasia grade 3 (CIN3).

## Data Availability

The datasets analyzed during the current study are not publicly available because these materials also form part of an ongoing study, but are available from the corresponding author on reasonable request.

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
