# Peer review of "Absolute Risk and Attributable Fraction of Type-Specific Human Papillomavirus in Cervical Cancer and Precancerous Lesions—A Population-Based Study of 6286 Women in Rural Areas of China"

_jcm, 2022, doi:10.3390/jcm11216483_

Round 1

Reviewer 1 Report

This is an interesting HPV cross-sectional prevalence study located in Inner Mongolia and Shanxi province, China. They highlight strains of HPV that are not currently targeted by the vaccine but that could be included in future HPV vaccines. The methods are thorough and clearly presented. The results are dry but the topic is of interest as we formulate more HPV vaccines to prevent cervical cancer. The discussion puts the findings into an appropriate context. Overall well done and relevant study. See minor points below.

Line 148—the same equation/numbers are brought up to calculate the 1.8 and 0.2 attributions in the HPV AF explanation. Please explain where then numbers come from.

Line 159—there is a strange symbol used repeatedly in presenting these statistics, eg “twirly line” 2 = 5.20, please explain what is meant by this or replace with the appropriate abbreviation throughout the manuscript

Table 1—please consider replacing “drinking” with “alcohol use” or another form that clarifies its association with alcohol.

In the table2-5, please clarify that the data are presented as number (percentage). (eg add N(%) to the title or elsewhere in the table)

Line 240—it would also be interesting to see what HPV strains are present in women who have had the HPV vaccination, to see what breakthrough infections exist

Line 250-251 Should be presented in the results

Lines 333—I would add limitations for risk factors in that it was asked of patients, which may or may not be accurate for that portion of the analysis. Consider also addressing that smoking is not a risk factor in this study, but that it has repeatedly been shown to be associated with HPV infection and cervical cancer—is it that the sample of women who smoked is too small?

Author Response

This is an interesting HPV cross-sectional prevalence study located in Inner Mongolia and Shanxi province, China. They highlight strains of HPV that are not currently targeted by the vaccine but that could be included in future HPV vaccines. The methods are thorough and clearly presented. The results are dry but the topic is of interest as we formulate more HPV vaccines to prevent cervical cancer. The discussion puts the findings into an appropriate context. Overall well done and relevant study. See minor points below.

Thank you very much for your suggestions! I have revised the manuscription carefully according to the advices and the detailed corrections are shown below.

(1) Line 148—the same equation/numbers are brought up to calculate the 1.8 and 0.2 attributions in the HPV AF explanation. Please explain where then numbers come from.

Answer: The calculation was conducted according to reference 18 (Insinga RP et al. 2008), which was added (Line 148).

(2) Line 159—there is a strange symbol used repeatedly in presenting these statistics, eg “twirly line” 2 = 5.20, please explain what is meant by this or replace with the appropriate abbreviation throughout the manuscript

Answer: This is format issue. The correct expression is c2=5.20, the chi square symbol “c” was lost.

(3) Table 1—please consider replacing “drinking” with “alcohol use” or another form that clarifies its association with alcohol.

Answer: I replaced “drinking” with “alcohol use”.

(4) In the table2-5, please clarify that the data are presented as number (percentage). (eg add N(%) to the title or elsewhere in the table)

Answer: Table 2 to table 5 were updated according to your suggestion.

(5) Line 240—it would also be interesting to see what HPV strains are present in women who have had the HPV vaccination, to see what breakthrough infections exist

Answer: Since we started to enroll subjects in 2016 when the first HPV vaccination was officially approved in China, the data is not available in this study.

(6) Line 250-251 Should be presented in the results

Answer: The data was added in Line 250 was shown in Line 201, and data in Line 251 was shown in Line 222.

(7) Lines 333—I would add limitations for risk factors in that it was asked of patients, which may or may not be accurate for that portion of the analysis. Consider also addressing that smoking is not a risk factor in this study, but that it has repeatedly been shown to be associated with HPV infection and cervical cancer—is it that the sample of women who smoked is too small?

Answer: Although other studies reported smoking as a risk factor, the smoking rate in this study is too small to evaluate its role in HPV infection as your assumption. This -limitation was added (Line347-348).

Reviewer 2 Report

Absolute risk and attributable fraction of type-specific HPV in cervical cancer and precancerous lesions – a population-base study of 6286 women in rural areas of China by Li et al.

In this cross-sectional study, the authors present the AR/AF/RR of cervical HPV infection and lesions among women from Inner Mongolia and Shanxi in China. The manuscript itself is interesting and of significant general interest. Both the population examined, and the results are quite unique. However, the presentation of the results obscures rather than highlights the interesting findings. Additionally, editorial assistance from a native English speaker should be sought prior to publishing.   Below are a few suggestions for the author’s consideration:

Abstract

Line 18: could also describe % of protection for HPV 16 and 18 to be consistent with the AF described in the prior sentence

Introduction

-line 43: please add a reference for this sentence

-The introduction paragraphs 2 and 3 probably could be compressed by narrowing the descriptive data to local or regional information and presenting the information more succinctly

-lines 77-78: please add a reference for this sentence

-the study hypothesis in the introduction is vague and needs to be described more clearly. I recommend omitting the words “real danger” in line 84

Material and Methods

-HPV tests: it seems like the first aliquoted PreservCyt medium was aliquoted and Cobas or Aptima tested – how did you decide which assay will be used for HPV detection?

-Colposcopy: is it routine to obtain random biopsies from the cervix when you do not see any lesions on visual inspection? How do you decide on the biopsy site – is it standardized?

-I think the disease status classification is a bit confusing. Maybe group 1 would be the group with “Negative high-grade Intraepithelial lesion” when there is no histological evidence of CIN à by default, the group that will get to the colposcopic exam will be those with HPV DNA detection and ascus or more. The other group would be persons with “positive high-grade intraepithelial lesion” by either cytology or colposcopy. You could consider excluding the unsatisfactory because we don’t know whether they have abnormal cells or not. Also, I don’t see using this classification in the result section

Statistical analysis

-I don’t see the group of HPV vaccine types present in the nonavalent HPV vaccine (Gardasil-9)-  you mentioned in your introduction that the nonavalent HPV vaccine is available in China.

-It is not clear to me why you distinguish between single type vs co-infection with multiple types in the AR but not in the AF

-line 15 (and also in the results): there is a symbol near the word “Pearson’s” that is not standard

-In paragraph 2 of the results, it would be clearer and easier to follow if you first describe the prevalence of cytology findings and then pathology findings. And then describe the HPV findings according to cytology/pathology results

-Table 1: needs footnotes about how age was measured (e.g., “years”); what HPV types are included in the “HPV Prevalence” group described in the 3rd column; what types of contraception are included in the “others” category

-In section 3.3, the results presented are not well pre-specified in the methods section. 

-I will invert the order of tables 2 and 3 (cytology first and then histology). I think if you keep the order: HPV DNA, then cytology, then histology would be easier to follow

-Results of tables 4 and 5 should be presented before the results of tables 2 and 3

-since you have the data, could also present how many people with Hr-HPV and abnormal cytology ended up having histologically-proven high-grade lesions 

Discussion

-similar to the results section, would start in the same order (HPV DNA, then cytology, then histology)

-No need to repeat numbers from the result section in the conclusion section; this will help you to provide a more succinct discussion

-like 257: could you please provide some examples of some “unmeasured high-risk behaviors or habits” in this population?

-Lines 258-261: since you are describing that testing with different assays could affect your results, please briefly provide information about the different assays perform

-4th paragraph of the discussion: not sure if I understand the comment in the last 2 sentences; please review for clarity

-5th paragraph of the discussion is excellent but could be more succinct

-6th (limitation) paragraph, you mention that none of the women underwent cervical cancer screening in the past three years – I do not recall that in your inclusion criteria or Table 1. Please make sure is mentioned in your methods or results and not just in the discussion

-I am not sure what “high-level doctors” mean, is that “trained or certified providers” to  read the pathology

Author Response

Thank you very much for your suggestions! I have revised the manuscription carefully according to the advices and the detailed corrections are shown below.

Abstract

(1) Line 18: could also describe % of protection for HPV 16 and 18 to be consistent with the AF described in the prior sentence

Answer: I revised the abstract (Line30) according your suggestion.

Introduction

(2) line 43: please add a reference for this sentence

Answer: The reference was added.

(3) The introduction paragraphs 2 and 3 probably could be compressed by narrowing the descriptive data to local or regional information and presenting the information more succinctly

Answer: These paragraphs were revised.

(4) lines 77-78: please add a reference for this sentence

Answer: The reference was added. The data comes from Chinese Drug Clinical Trial Registration and Information Publicity Platform (http://www.chinadrugtrials.org.cn/index.html), which belong to Chinese Center for Drug Evaluation.

(5) the study hypothesis in the introduction is vague and needs to be described more clearly. I recommend omitting the words “real danger” in line 84

Answer: This part was revised (Line 84).

Material and Methods

(6) HPV tests: it seems like the first aliquoted PreservCyt medium was aliquoted and Cobas or Aptima tested – how did you decide which assay will be used for HPV detection?

Answer: The samples from Shanxi Province were tested by Aptima and samples from Inner Mongolia were tested by Cobas.

(7) Colposcopy: is it routine to obtain random biopsies from the cervix when you do not see any lesions on visual inspection? How do you decide on the biopsy site – is it standardized?

Answer: Yes, it is standardized, 4-quadrant punch biopsy (Squamocolumnar junction at 2,6,8,12 o’clock position) and endocervical curettage were conducted if no visual lesions were detected for the subjects with cytology LSIL+.

(8) I think the disease status classification is a bit confusing. Maybe group 1 would be the group with “Negative high-grade Intraepithelial lesion” when there is no histological evidence of CIN à by default, the group that will get to the colposcopic exam will be those with HPV DNA detection and ascus or more. The other group would be persons with “positive high-grade intraepithelial lesion” by either cytology or colposcopy. You could consider excluding the unsatisfactory because we don’t know whether they have abnormal cells or not. Also, I don’t see using this classification in the result section

Answer: Thanks a lot for your suggestion. To make it clear, I revised it in the results (Line209) and table 2 – table5 (replaced “normal” with “NILM” in Table 2 and table 4, replaced “normal” with “Negative” table3 and table5).

.

Statistical analysis

(9) I don’t see the group of HPV vaccine types present in the nonavalent HPV vaccine (Gardasil-9)- you mentioned in your introduction that the nonavalent HPV vaccine is available in China.

Answer: Nine valent HPV vaccine (Gardasil-9) group were shown in line 138-139 HPV 6/11/16/18/31/33/45/52/58.

(10) It is not clear to me why you distinguish between single type vs co-infection with multiple types in the AR but not in the AF.

Answer: According to the formula, absolute risk (AR) is the infection rate of HPV that does not distinguish multiple infection from single infection, it is hard to evaluate the role of each HPV type; AF[20] is calculated on the relative number of instances in which each HPV infection type was observed as a single-type infection in a lesion of that grade and disease-type within each individual study, which is more accurate than AR.

[20]. Zhao XL, Hu SY, Qian Z, Li D, Feng RM, Han R, Zhao FH: High-risk human papillomavirus genotype distribution and attribution to cervical cancer and precancerous lesions in a rural Chinese population. Journal of Gynecologic Oncology 2017, 28(4):e30.

(11) line 15 (and also in the results): there is a symbol near the word “Pearson’s” that is not standard

Answer: This is format issue. It means “Pearson’s c2”, the symbol was lost at format conversion process.

(12) In paragraph 2 of the results, it would be clearer and easier to follow if you first describe the prevalence of cytology findings and then pathology findings. And then describe the HPV findings according to cytology/pathology results

Answer: Paragraph 2 in the results shows the HPV infection rate, cytology distribution and pathology distribution in different age groups (Fig1). It might be appropriate to keep it in the section of demographic characteristics and HPV distribution.

(13) Table 1: needs footnotes about how age was measured (e.g., “years”); what HPV types are included in the “HPV Prevalence” group described in the 3rd column; what types of contraception are included in the “others” category

Answer: The related footnotes were added in Table1

.

(14) In section 3.3, the results presented are not well pre-specified in the methods section.

Answer: All of the HPV grouping and classification of cytology and pathology were pre-specified in the methods section (Line 110-140).

(15) I will invert the order of tables 2 and 3 (cytology first and then histology). I think if you keep the order: HPV DNA, then cytology, then histology would be easier to follow

Answer: Yes, we did put cytology first and then histology as your suggestion. In the manuscript, Table 2 showed AR and AF in cytology, and table 3 in histology. The title of table 2 and 3 were changed to avoid the misunderstanding.

(16) Results of tables 4 and 5 should be presented before the results of tables 2 and 3

Answer: From the structure of the paper, it showed AR and AF of each HPV type in cytology and histology first, and the protection rate was calculated for the different combination of HPV types according to HPV vaccines. We think it would be clearer and easier to understand.

(17) since you have the data, could also present how many people with HR-HPV and abnormal cytology ended up having histologically-proven high-grade lesions.

Answer: Since not all the subjects of this study went through colposcopy, the final disease status for the women with no pathology results were grouped into different pathology grades as below [17, 18]: (1) Negative pathological diagnosis: a) colposcopy results negative, b) HPV DNA test negative and LBC diagnosed NILM or ASCUS, c) HPV DNA test positive and LBC diagnosed NILM. (2) CIN2 group: cytology diagnosis was ASC-H, HSIL, AIS or SCC. (3) no pathological results group: a) HPV negative and LBC diagnosed Unsatisfactory, LSIL or AGC, b) HPV positive and LBC diagnosed ASCUS or LSIL(Line128-Line134). So we can’t get the data for people with HR-HPV and abnormal cytology ended up having histologically-proven high-grade lesions. Therefore, it is hard to generate these figures.

[17].    Belinson JL, Qiao YL, Pretorius RG, Zhang WH, Rong SD, Huang MN, Zhao FH, Wu LY, Ren SD, Huang RD et al: Shanxi Province cervical cancer screening study II: self-sampling for high-risk human papillomavirus compared to direct sampling for human papillomavirus and liquid based cervical cytology. International journal of gynecological cancer : official journal of the International Gynecological Cancer Society 2003, 13(6):819-826.

[18].    Zhao FH, Hu SY, Zhang Q, Zhang X, Pan QJ, Zhang WH, Gage JC, Wentzensen N, Castle PE, Qiao YL: Risk assessment to guide cervical screening strategies in a large Chinese population. International Journal of Cancer Journal International Du Cancer 2016, 138(11):2639-2647.

Discussion

(18) similar to the results section, would start in the same order (HPV DNA, then cytology, then histology)

Answer: See the answer to the question 16.

,

(19) No need to repeat numbers from the result section in the conclusion section; this will help you to provide a more succinct discussion

Answer: The description was simplified (Line 256-257, Line 274-275).

(20) like 257: could you please provide some examples of some “unmeasured high-risk behaviors or habits” in this population?

Answer: It means other contraception measures, e.g. coitus interruptus and fertile period contraception.

(21) Lines 258-261: since you are describing that testing with different assays could affect your results, please briefly provide information about the different assays perform

Answer: The sensitivity of cobas and Aptima was adapted according to ROC area for CIN2+ detection. The cutoff value ranged from 100 to 1000 copies/ml. SPF10 is a PCR-based reverse hybridization, which can detect much lower level of HPV than cobas and Aptima. Routinely, cobas and Aptima are used for cervical cancer screening. In contrast, SPF10 line blot is used for HPV infection monitoring and epidemiology study, which is better for the evaluation of AR and AF of HPV types.

(22) 4th paragraph of the discussion: not sure if I understand the comment in the last 2 sentences; please review for clarity

Answer: It was revised in the manuscript: Considering the importance of 52 and 58 in CIN2+, future HPV based cervical cancer screening technology may need to genotype these two strains (Line302-303).

(23) 5th paragraph of the discussion is excellent but could be more succinct

Answer: The 5th paragraph was simplified.

(24) 6th (limitation) paragraph, you mention that none of the women underwent cervical cancer screening in the past three years – I do not recall that in your inclusion criteria or Table 1. Please make sure is mentioned in your methods or results and not just in the discussion

Answer: It was added in the methods of eligibility criteria in Line 94 to 95.

(25) I am not sure what “high-level doctors” mean, is that “trained or certified providers” to read the pathology

Answer: It means senior doctor, and we corrected it in Line 342.